# Modelling Potential Distribution of Snow Leopards in Pamir, Northern Pakistan: Implications for Human–Snow Leopard Conflicts

**Wajid Rashid** [1,2], **Jianbin Shi** [1,*], **Inam ur Rahim** [3], **Muhammad Qasim** [2], **Muhammad Naveed Baloch** [4], **Eve Bohnett** [5], **Fangyuan Yang** [4], **Imran Khan** [6] and **Bilal Ahmad** [7]

1. School of Environment, Beijing Normal University, Beijing 100875, China; wajid@uswat.edu.pk
2. Department of Environmental and Conservation Sciences, University of Swat, Mingora 19130, Pakistan; drqasim@uswat.edu.pk
3. Centre for Applied Policy Research in Livestock (CAPRIL), Department of Climate Change and Livestock, University of Veterinary and Animal Sciences, Lahore 54600, Pakistan; inam.rahim@uvas.edu.pk
4. Key Laboratory of Zoological Systematics and Evolution, Institute of Zoology, Chinese Academy of Sciences, Beijing 100101, China; naveedmalkani@ioz.ac.cn (M.N.B.); yangfy@ioz.ac.cn (F.Y.)
5. Department of Biology, San Diego State University, San Diego, CA 92182, USA; ebohnett@sdsu.edu
6. State Key Laboratory of Animal Nutrition, Institute of Animal Sciences, Chinese Academy of Agricultural Sciences, Beijing 100193, China; imran_talash76@yahoo.com
7. Institute of Agriculture Sciences and Forestry, University of Swat, Mingora 19130, Pakistan; bilalforester89@yahoo.com
* Correspondence: jbshi@bnu.edu.cn

**Abstract:** The snow leopard (*Panthera uncia*) is a cryptic and rare big cat inhabiting Asia's remote and harsh elevated areas. Its population has decreased across the globe for various reasons, including human–snow leopard conflicts (HSCs). Understanding the snow leopard's distribution range and habitat interactions with human/livestock is essential for understanding the ecological context in which HSCs occur and thus gives insights into how to mitigate HSCs. In this study, a MaxEnt model predicted the snow leopard's potential distribution and analyzed the land use/cover to determine the habitat interactions of snow leopards with human/livestock in Karakoram–Pamir, northern Pakistan. The results indicated an excellent model performance for predicting the species' potential distribution. The variables with higher contributions to the model were the mean diurnal temperature range (51.7%), annual temperature range (18.5%), aspect (14.2%), and land cover (6.9%). The model predicted approximately 10% of the study area as a highly suitable habitat for snow leopards. Appropriate areas included those at an altitude ranging from 2721 to 4825 m, with a mean elevation of 3796.9 ± 432 m, overlapping between suitable snow leopard habitats and human presence. The human encroachment (human settlements and agriculture) in suitable snow leopard habitat increased by 115% between 2008 and 2018. Increasing encroachment and a clear overlap between snow leopard suitable habitat and human activities, signs of growing competition between wildlife and human/livestock for limited rangeland resources, may have contributed to increasing HSCs. A sound land use plan is needed to minimize overlaps between suitable snow leopard habitat and human presence to mitigate HSCs in the long run.

**Keywords:** habitat fragmentation; habitat suitability; land use/cover change; *Panthera uncia*; MaxEnt model

## 1. Introduction

The snow leopard (*Panthera uncia*) is a rare and secretive carnivore inhabiting the high Asian mountainous regions of 12 Asian countries [1]. It is a flagship and umbrella species that has attained an iconic status and conservation priority around the globe [2], mainly because the conservation of snow leopards has direct implications for the conservation of

the overall biodiversity in these high Asian regions [3,4]. Protecting snow leopards will correspondingly help conserve these high-altitude rangeland and wetland ecosystems. These ecosystems can in turn provide necessary ecosystem services required for human wellbeing and sustainable development [5–7].

During the last few decades, the population of snow leopards has been declining steadily in its global range [8–10]. The main reasons for this are the increasing human population, habitat fragmentation, illegal hunting, declining populations of their prey species, and livestock-depredation-induced human–snow leopard conflicts (HSC) that usually lead to retaliatory killings of snow leopards [1,9,11,12]. At times, snow leopards, similar to other large carnivores, are perceived as a real threat to livestock and trophy ungulates [13–15]. The decline of an apex carnivore, such as the snow leopard population, has an enormous impact on the food web and the habitat [16,17]. The declining population of snow leopards is also a threat to the ecosystem services these high-altitude regions provide [18–20].

The snow leopard was listed in the IUCN (International Union for Conservation of Nature) red list as an endangered species for 45 years and recently was downgraded to vulnerable status [21,22]. However, the conservation community debates this change in its conservation status from endangered to vulnerable [23,24]. Some conservation scholars believe that only a tiny percentage of the snow leopard global range has been surveyed with reliable scientific methods [25]. Suryawanshi et al. [26] found that snow leopard population assessment occurred in only 6 out of the 12 range countries. Researchers have surveyed snow leopard populations within 0.3–0.9% of the global snow leopard range [27]. However, with the limited knowledge currently available, the population of snow leopards is projected to decline by at least 10% in the next 22.62 years (or 3 generations) [21,27]. Researchers estimate that the global snow leopard range has contracted by almost 69% from its previous range a few decades ago [10].

The northern mountainous provinces of Pakistan confine the snow leopard to high elevation habitats within Gilgit Baltistan, Khyber Pakhtunkhwa, and Azad Kashmir [28]. The snow leopard range encompasses the great mountain ranges of c. 80,000 km$^2$, including the Himalayas, the Karakorum, the Hindukush, and Pamir [11,28–30]. These mountain ranges are a natural divide between South Asia and Central Asia [30]. Previous studies have estimated the number of snow leopards in Pakistan to be approximately 250 individuals sparsely distributed throughout a wide range [13,28]. Though the density of snow leopards is low, HSCs are prevalent, mainly due to livestock depredation in the Karakoram region of northern Pakistan ([31] our unpublished data). Snow leopards are then retaliatorily killed by livestock owners, which constitutes a significant threat to the species' long-term survival in these regions.

Species Distribution models (SDMs) can determine the relationship between records of species occurrences at known locations and the environmental characteristics of these locations [32,33]. If both presence and absence data are available, statistical methods are available to model species distributions [34]. However, it is rare to obtain true absence data in field ecological studies, and this limitation has led to the development of presence-only modelling approaches. MaxEnt, the maximum entropy machine learning algorithm, is one such approach [35]. MaxEnt is one of the most widely used SDMs for predicting the potential distribution of a given species [36–38]. MaxEnt uses presence-only data and eco-geographical variables (EGV) to model the habitat suitability for a given species [39–41]. A previous study by Fonderflick, et al. [42] reveals that the MaxEnt model is very suitable for elusive species for which absence data are unavailable or unreliable due to their rarity and thin density in their ranges. MaxEnt has successfully predicted the distribution range of snow leopards in different countries, including China, India, Nepal, and Kazakhstan [43–46]. However, no previous study has researched the core of known snow leopard distribution in Pakistan. Furthering our knowledge of the snow leopard distribution range and habitat interactions with humans is essential for understanding the

ecological context in which HSCs occur and for strengthening attempts to manage them in this region.

The current study aimed to model the snow leopard distribution in the core of its known distribution in northern Pakistan and assess the overlap of the snow leopard habitat with human activities. Specifically, the purposes of this study were to: (a) predict the distribution range of snow leopards in Northern Pakistan using a MaxEnt model; (b) determine the effect of eco-geographical variables on the snow leopard distribution and habitat suitability at various spatial scales; and (c) identify overlapping areas between the snow leopard distribution range and areas used by humans, which has implications for the management of HSCs.

## 2. Materials and Methods

### 2.1. Study Area

The study area lies inside the Khunjerab National Park (KNP) and valleys in KNP's buffer zone (the Khunjerab and Shimshal valleys) in the Gojal subdistrict of the Gilgit Baltistan province, encompassing approximately 8582 km$^2$ (36°33' N to 37°01' N; 74°52' E to 76°02' E) [40,47] (Figure 1). The altitude of the study area varies from 2439 m to 7885 m above sea level [40,48]. Precipitations range from 200 to 900 mm per annum, and most precipitation is in the form of snow in winter. Average temperatures range from below 0 °C from October onward and rise to approximately 27 °C in May [49].

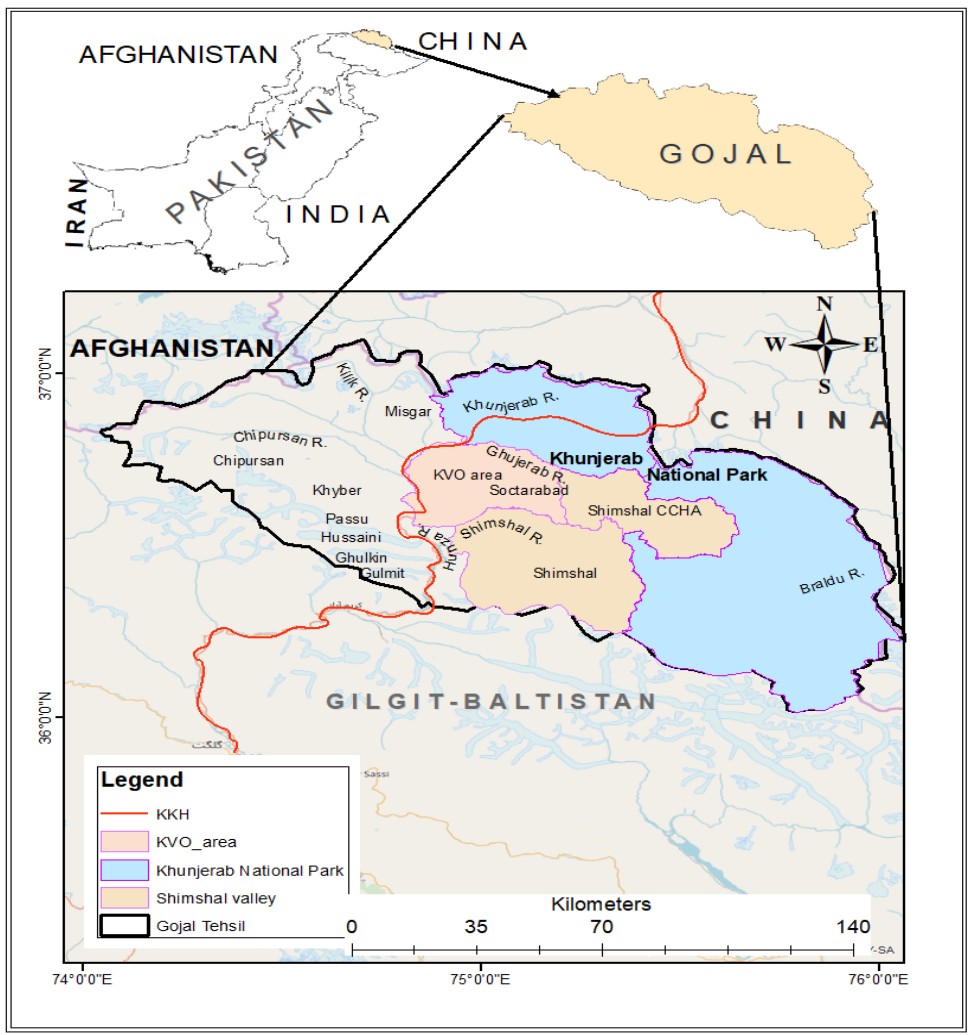

**Figure 1.** Map of the study area (coloured) in Gojal sub-district, Hunza, northern Pakistan.

The key mammalian species in the study area are snow leopard, grey wolf (*Canis lupus*), brown bear (*Ursus arctos isabellinus*), Eurasian lynx (*Felis lynx*), Tibetan red fox (*Vulpes vulpes montana*), Siberian ibex (*Capra sibirica*), blue sheep (*Pseudois nayaur*), Marco polo sheep (*Ovis ammon polii*), golden marmot (*Marmota caudata aurea*) and Cape hare (*Lepus capensis*) [40].

## 2.2. Occurrence Data

Field observations from sign surveys and camera trapping data were collected from March to May 2018, comprising the presence data of the snow leopard. The study area was divided into 5 × 5 km grids, selecting random grid cells for the snow leopard presence signs (including pug marks, scraps, scats, spray, or scent marks). The surveys identified snow leopard signs within a 50 m buffer area around each sampling point within each grid.

This study set a total of 30 cameras (Model Ltl Acorn-6210 M, Desa Moineis, IA, USA) for 20 days. A bait in the form of fish oil was placed at the camera trapping stations to increase the snow leopards' detectability. The camera traps were fully automatic and kept running to take three photos at each trigger for the study period as the animal density was low there. The camera trap locations within the grids were carefully selected to cover all altitudinal ranges and habitat types.

The presence signs were recorded with a Global Positioning System (GPS) and explicitly documented on a prepared document sheet. We recorded 104 signs of snow leopard presence in the study area and, after removing the duplicates signs located at a short distance (to avoid over-estimation), retained a final total of 85 records (Supplementary Table S2).

ArcGIS software (Version 10.7, 380 New York Street Redlands, CA, USA) was used for the preliminary data processing and extracting (clipping) images for our study area. ArcGIS was used to convert the snow leopard occurrence points to a raster format, followed by converting the downloaded eco-environmental variables into the ASCII format required for the MaxEnt Software (American Museum of Natural History, NY, USA).

## 2.3. Variables for the MaxEnt Model

We collected 25 variables, including 19 bio-climatic variables and 6 other variables, as predictors for the MaxEnt Model (Table 1). The 19 bio-climate variables were obtained from the website (http://www.worldclim.org) (accessed on 19 January 2020) with a spatial resolution of 30 Arc Second (approximately 1 km) [50–52]. The Digital Elevation Model (DEM) derived five variables, including terrain ruggedness index (TRI), slope, elevation, aspect, and distance to water, using ArcGIS 10.7. The program calculates TRI using the elevation difference among the neighboring cells in the DEM [53]. The land cover data were obtained from the Global Land Cover (2009) (http://due.esrin.esa.int/page_globcover.php) (accessed on 2 February 2020) processed by the European Space Agency and the Université Catholique de Louvain [45,50].

SPSS (IBM SPSS Statistic 26, Armonk, NY, USA) was used to perform Principal Component Analysis (PCA) and correlation analysis (Supplementary Table S1) to remove the independent variables that were highly correlated (correlation coefficient $|r| > 0.7$) [3,53,54]. The correlation test removed highly correlated independent variables and retained the independent variables with high explanatory power for the snow leopard presence [53,55,56].

The 10 final environmental variables (indicated by bold letters in Table 1) retained for use in the MaxEnt model, along with the 85 occurrence records, were used to predict the relationship between the snow leopard habitat distribution and the present bio-climatic conditions. The predictions show the relationship between snow leopard presence data and the environmental variables defined in the study area. The format of the 10 environmental variables was changed from raster to ASCII format in Arc GIS 10.7.

**Table 1.** List of the variables obtained and used for the MaxEnt Model.

| Variable Code | Variable Definition | Unit |
|---|---|---|
| Bio 1 | Annual mean temperature | °C |
| **Bio 2** | Mean diurnal range: mean of monthly (max temp−min temp) | °C |
| Bio 3 | Isothermality (Bio 2/Bio 7) (×100) | % |
| **Bio 4** | Temperature seasonality (standard deviation × 100) | °C |
| Bio 5 | Maximum temperature of warmest month | °C |
| Bio 6 | Minimum temperature of coldest month | °C |
| **Bio 7** | Temperature annual range (Bio 5−Bio 6) | °C |
| **Bio 8** | Mean temperature of wettest quarter | °C |
| Bio 9 | Mean temperature of driest quarter | °C |
| Bio 10 | Mean temperature of warmest quarter | °C |
| Bio 11 | Mean temperature of coldest quarter | °C |
| Bio 12 | Annual precipitation | mm |
| **Bio 13** | Precipitation of wettest month | mm |
| Bio 14 | Precipitation of driest month | mm |
| **Bio 15** | Precipitation seasonality (coefficient of variation) | % |
| **Bio 16** | Precipitation of wettest quarter | mm |
| Bio 17 | Precipitation of driest quarter | mm |
| Bio 18 | Precipitation of warmest quarter | mm |
| Bio 19 | Precipitation of coldest quarter | mm |
| Ruggedness | Elevation difference between adjacent cells | m |
| DEM | Altitude | m |
| **Slope** | Slope angle | Degree |
| Dist_water | Distance to Water | m |
| **Land cover** | Land cover type (Categorical) | |
| **Aspect** | Direction of slope | Degree |

Note. The environmental variables in bold were selected and retained (using correlation coefficient $|r| > 0.7$) after eliminating the highly correlated variables for constructing the MaxEnt model.

### 2.4. Land Use/Land Cover (LULC) Dynamics

This study developed a land use/cover map to reflect human activity interventions from 2008 to 2018 to assess the impact of land cover changes on snow leopard habitat. The land use/cover methods classified the Landsat 8 data products, freely accessible and available on the USGS (United States Geological Survey) website (https://earthexplorer.usgs.gov/, accessed on 26 February 2020). The images were processed using GIS application tools to develop land cover maps from 2008 through 2018 to detect land cover changes in the study area. We first performed atmospheric correction on the images using Radiometric Calibration and FLAASH Atmospheric Correction functions in ENVI 5.3 (Broomfield, CO, USA). Second, the Normalized Differential Vegetation Index (NDVI) and Modified Normalized Difference Water Index (MNDWI) were calculated using Raster Calculator in ArcGIS, with the formula "(Band5−Band4)/(Band5+Band4)" and formula "(Band2−Band5)/(Band2+Band5)", respectively. Then, the landcover data were obtained by interactive supervised classification to the image after compositing of band 4, 5 and 6, using Image Classification in ArcGIS. According to the image color and field facts, we specified the following six categories in the training set: snow/glacier, barren land, rangeland, agriculture, built-up area, and water bodies. Finally, we counted the land-use changes between 2008 and 2018 in ArcGIS.

### 2.5. MaxEnt Distribution Model

MaxEnt software generates the probability of a species presence. The MaxEnt software (Version 3.4.1) is available from http://biodiversityinformatics.amnh.org/open_source/maxent (accessed on 19 January 2020). The training data represented 80% of the sample data and were randomly selected, whereas the remaining 20% were the test data [57,58].

The model included the following settings. A logistic output format described the probability of snow leopard presence, with continuous habitat suitability ranging from 0 (unsuitable) to 1 (most suitable) [53]. MaxEnt requires a threshold value to distinguish suit-

able from unsuitable habitats [34,59,60]. This study used the maximum sum of sensitivity and specificity (Max SSS) to provide the threshold value, as recommended by previous studies [54,61].

The program performed 20 model iterations to generate an average result. The random test percentage was 20, the regularization multiplier was 1, and the maximum number of iterations of background points was 10,000 [36]. The convergence threshold was equal to 0.00001, and the replicates type was subsample.

The Jackknife test result showed the accuracy gained from a single variable or a combination of all variables. The greater the value gained, the more information or contribution the variable made to habitat distribution [55]. Model evaluation for SDM performance used the Area Under the Receiver Operating Characteristic curve (AUC) [57]. The AUC and the performance of the prediction model were positively correlated [62]. In the current study, the average AUC based on the 20 replicates provided a criterion for model performance. Generally, the value of AUC ranges from 0.5 to 1. An AUC equal to 0.5 means the model performance is equal to pure guessing. Based on the guidelines by Swets [63], the grading of the model performance ranges from fail (0.5 to 0.6), poor (0.6 to 0.7), fair (0.7 to 0.8), good (0.8 to 0.9) to excellent (0.9 to 1) [35,51,64].

## 3. Results

### 3.1. Analysis of the Variable Contributions

The variables with the highest contributions in the MaxEnt model were the mean diurnal temperature range (Bio 2, 51.7%), the annual temperature range (Bio 7, 18.5%), the aspect or direction of the slope (14.2%), and land cover (6.9%) (Table 2).

**Table 2.** Percent contribution and permutation importance values of each environmental variable in the snow leopard habitat suitability model in decreasing order of contribution.

| Variables | Percent Contribution | Permutation Importance |
|---|---|---|
| Mean diurnal temperature range | 51.7 | 88.1 |
| Temperature annual range | 18.5 | 0.3 |
| Aspect | 14.2 | 0.7 |
| Land cover | 6.9 | 1.7 |
| Slope | 2.2 | 0.5 |
| Temperature seasonality | 2.0 | 0.7 |
| Precipitation of wettest month | 2.0 | 3.7 |
| Mean temperature of wettest quarter | 1.2 | 1.9 |
| Precipitation seasonality | 0.8 | 0.3 |
| Precipitation of wettest quarter | 0.4 | 2.0 |

Note: Refer to Table 1 for definitions of the variables.

### 3.2. Predicted Snow Leopard Habitat

The detection of snow leopards was low, as only 3 out of the 30 camera traps captured snow leopard photographs. The camera trapping stations detected snow leopards at lower elevations near the road (from 2990 m to 4140 m a.s.l.). However, the snow leopard prey species, including Ibex, Cape hare, and other small mammals, were mainly detected in the camera trapping stations at higher altitudes.

The suitable habitat for snow leopards in this region ranged from unsuitable (0) to suitable (1). The average test AUC for the replicate runs was excellent at 0.914, and the standard deviation was 0.046 (Supplementary Figure S4). MaxEnt predicted 859 km$^2$ (approximately 10%) of the study area as a highly suitable habitat for snow leopards. The study area's remaining 7723 km$^2$ (approximately 90%) was less suitable or even unsuitable for snow leopards in this region (Figure 2).

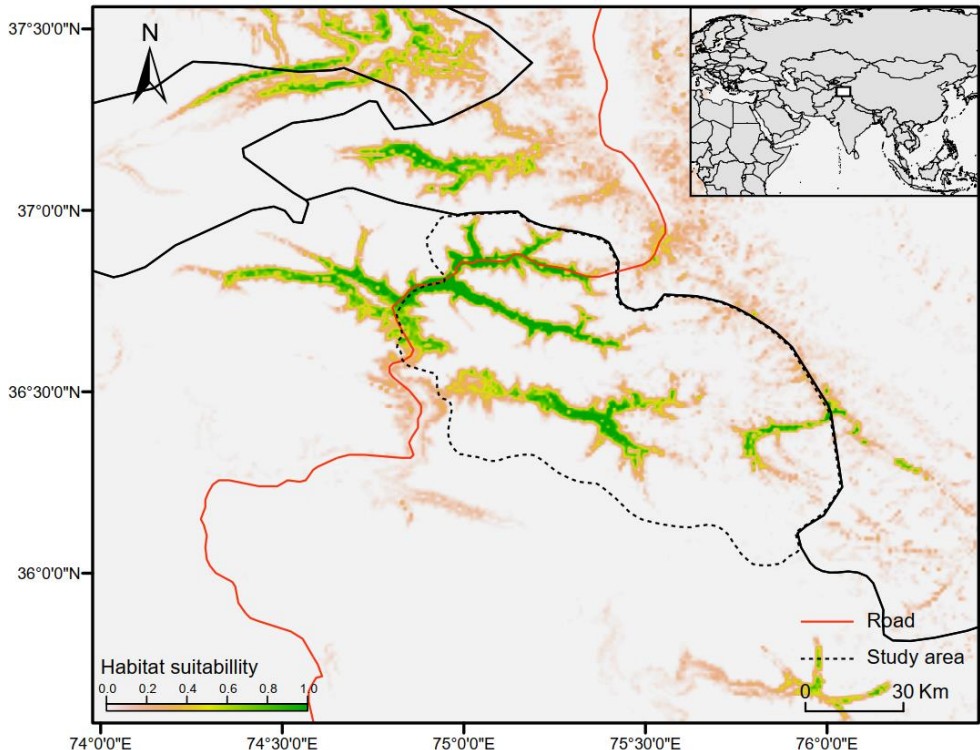

**Figure 2.** Suitable habitat identified for snow leopards in the study area. The road linking Pakistan to China passes through some of the most suitable habitats.

### 3.3. Overlap of Suitable Habitat with Human Activities

The models identified suitable habitats for snow leopards at elevations ranging from 2721 to 4825 m a.s.l., with a mean elevation of 3797 ± 432 m a.s.l. The suitable habitat for snow leopards at these elevation ranges overlapped with human settlements and activities (Figure 2).

Most of the study area comprised of barren land (59.6%) and glacier/snow (34.2%) (Supplementary Figure S5 and Table 3). In contrast, rangelands only covered a minimal area (5.5%), with wildlife such as snow leopard prey ungulates sharing the range with continually increasing number of livestock. Built-up and agricultural areas together accounted for less than 0.4% of the study area, but they increased sharply and overlapped with the suitable habitat for snow leopards.

**Table 3.** Land use and land cover changes from 2008 to 2018 in the study area.

| Land Use | 2008 | | 2018 | | Change in Area (ha) | Change in Percent (%) |
|---|---|---|---|---|---|---|
| | Area (ha) | % | Area (ha) | % | | |
| Barren | 403,750 | 47.05 | 511,110 | 59.56 | +107,360 | +26.59 |
| Snow/Glaciers | 417,354 | 48.63 | 293,900 | 34.25 | −123,454 | −29.58 |
| Rangeland | 33,470 | 3.90 | 47,200 | 5.50 | +13,730 | +41.02 |
| Water bodies | 2120 | 0.25 | 2720 | 0.32 | +600 | +28.30 |
| Built-up | 306 | 0.04 | 840 | 0.10 | +534 | +174.51 |
| Agriculture | 1200 | 0.14 | 2400 | 0.28 | +1200 | +100.00 |
| Total | 858,200 | 100 | 858,200 | 100 | | |

During the last 10 years (from 2008 to 2018), most of the human encroachment resulted in converting the most suitable habitat for snow leopards into agriculture and human settlements. Human settlements and agriculture covered 1506 ha in 2008 but increased by 115% to 3240 ha in 2018, a massive human encroachment into the most suitable habitat for snow leopards (Supplementary Figure S5).

## 4. Discussion

### 4.1. Predicted Snow Leopard Habitat

The current study determined the habitat suitability for snow leopards and their distribution in northern Pakistan using the MaxEnt model. It reveals that only 10% of the studied area in this region was highly suitable for snow leopards. MaxEnt has been increasingly employed to predict suitable habitats for snow leopards using occurrence records in global snow leopard ranges [22,43,46,65,66]. Similar to previous studies, only 12% of the distribution area is suitable for snow leopards in the neighbouring Ladakh region of Indian Administered Kashmir [45], and 11.6% of the study area is suitable habitat for snow leopards in Nepal [46]. In contrast, a recent study in Siberia (Russia) and adjacent areas (in China, Kazakhstan, and Mongolia) revealed a much smaller percentage (3.3%) of the study area as suitable snow leopard habitat [65]. In comparison, 22.7% of the total area is suitable habitat for snow leopards in the Qomolangma National Nature Reserve (QNNR) of Tibet (China) [67].

The study area mainly comprised very high elevations with rugged mountains, found to be unsuitable for and avoided by snow leopards. The suitable habitat for snow leopards lies at elevations ranging from approximately 2721 m to 4825 m, which is relatively low compared to the overall very high elevation (2439 m to 7885 m) in this study area. Holt et al. [43] reveals that the most suitable habitat for snow leopards lies at an elevation of 2500 m a. s. l. in Kazakhstan. A previous study on snow leopard habitat distribution found that the lower elevation limit of the snow leopard distribution follows a north–south gradient [65]. The lower elevation limit for snow leopards is 1220 m in Mongolia (in the northern snow leopard range) and 3350 m in Nepal (the southern parts of the Himalayas) [68]. In contrast, a study in the Sanjiangyuan region of the Tibetan plateau (China) showed that the suitable snow leopard habitat extends over 3334 m [44], but this could still be considered as lower elevation because the average elevation of this region is over 4500 m. A previous study showed that the Tibetan plateau is at the upper limit of the snow leopard range at 5180 m, while snow leopards prefer habitats at comparatively lower elevations of approximately 4000 m near Mount Everest (8844 m above sea level) [67]. Thus, it is evident that the suitable habitats for snow leopards lie at comparatively lower elevations across their global distribution ranges.

Previous studies have shown that an increasing human presence in wildlife habitats and resource use have increased human–wildlife conflicts in similar regions [69,70]. Suitable habitat for snow leopards in the lower elevations overlapped with the human settlements and human presence (Figure 3); resultant overlapping leads to human–snow leopard conflicts in this region. The expanding human population and infrastructure construction have been encroaching prime snow leopard habitats. The increasing human presence in the suitable habitats affects the snow leopard prey, causing human–snow leopard conflicts [71], thus threatening the survival of this emblematic rare carnivore in this region.

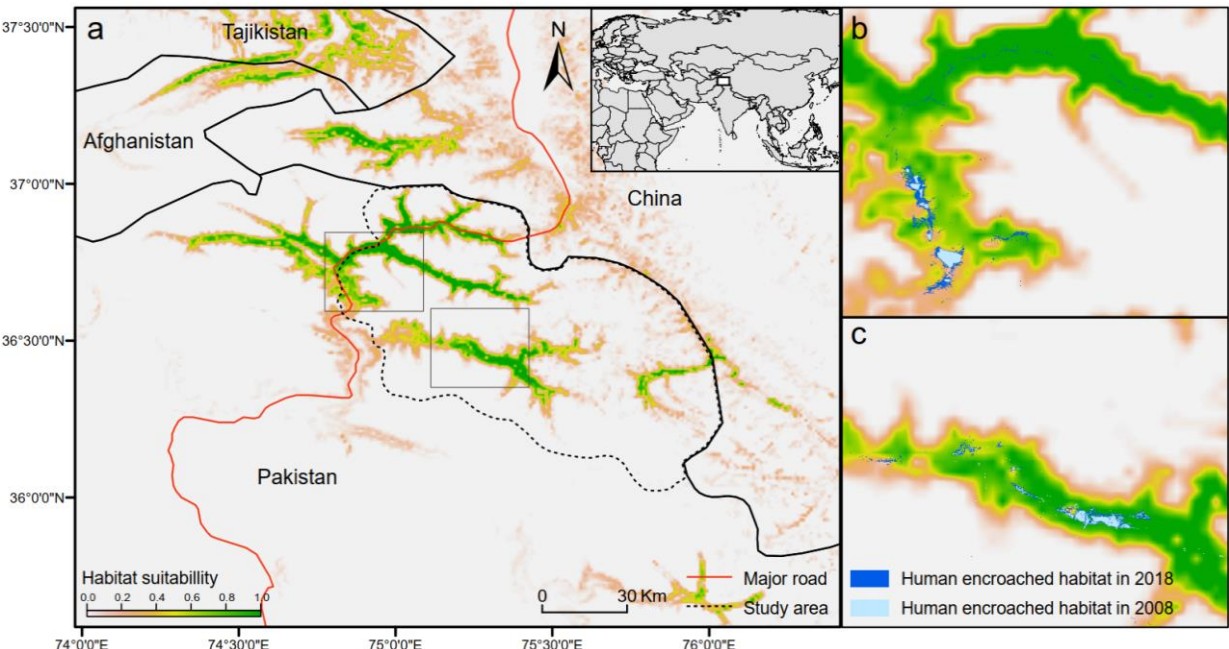

**Figure 3.** Maps showing: (**a**) Habitat suitability for the snow leopard in the study area; (**b**) Human encroachment in 2008 and 2018 (enlargement of the left upper rectangle in a); and (**c**) Human encroached habitat in 2008 and 2018 (enlargement of the lower right rectangle in (**a**)).

### 4.2. Factors Determining Suitable Habitats for Snow Leopards

Many different ecological and environmental factors determine habitat suitability. Our results showed that the mean diurnal temperature range (Bio2), the annual temperature range (Bio7), the aspect or direction of the slope, and land cover were the four main factors influencing habitat suitability of the snow leopard in the Karakoram–Pamir region of northern Pakistan. Singh et al. [72] obtained similar results predicting the snow leopard habitat suitability in the western mountains of India, revealing the mean diurnal temperature range (Bio2), slope, and aspect as primary factors in habitat suitability. The elevation, slope, land cover, and aspect determined habitat suitability for snow leopards in Kazakhstan [43], nearly identical to our results. In Nepal, the annual mean temperature (Bio1) is critical in predicting suitable habitats for snow leopards [46]. A study in India by Singh et al. [72] found that precipitation seasonality (Bio15) and precipitation of the wettest quarter (Bio16) were important factors for predicting habitat suitability for snow leopards which is identical to our results.

Contrary to our study, the Ladakh region of the Indian Administered Kashmir showed that elevation and ruggedness are the most important contributing factors to snow leopard distribution [45]. Another study in China concluded that precipitation in the driest quarter (Bio17), the ruggedness, the elevation, the maximum temperature of the warmest month (Bio5), and the annual mean temperature (Bio1) were the main contributing factors [67]. Cumulative precipitation in the winter and spring is the primary factor influencing the potential distribution of snow leopards in the Siberian region of Russia [65]. Similar to our results, the land use land cover is an essential factor for predicting the snow leopard habitat suitability in western India [72].

### 4.3. Land Use/Land Cover (LULC) Dynamics

The burgeoning human population, agriculture expansion, and infrastructure have enhanced the human presence in prime snow leopard habitats at lower elevations. The increasing expansion of agricultural activities and human settlement creates a conflict between humans and wildlife in many places [69]. The highly suitable habitat for the snow

leopards is increasingly encroached to meet this region's ever-growing human population and livestock needs.

Large areas of glacier/snow and barren land in our study area means that most of it is not suitable to support large populations of wild ungulates for snow leopards to prey on in this region. The minimal area of rangelands, shared by both wildlife and increasing number of domestic livestock, indicates ever-increasing competition between them for resources. Meanwhile, the rapid increase in urban and agricultural areas may bring more overlap with the suitable habitat for the snow leopards.

In addition to agriculture and human settlements expanding at lower elevations, in recent years, the number of tourists to this area has increased by 10,437.8% between 1999 and 2019 [73]. The increasing human population, domestic livestock, agricultural land, road infrastructure, and tourism-associated activities have fragmented the suitable habitats for snow leopards [22,74]. Such fragmentation can also bring the human/livestock and snow leopards into closer contact with each other, and the increasing proximity of large carnivores and human/domestic livestock is one of the main reasons for the depredation of livestock by snow leopards [70,75]. Depredation of livestock by snow leopards is, in turn, causing increasing HSCs in northern Pakistan and its global habitat [76].

Human density around a protected area is one of the main reasons for the local extinction of species, as in the case of wild ungulates and carnivores in Ghana (western Africa) [77]. An increasing number of houses around protected areas is related to a decreased ability of protected areas to conserve species [78]. The increased human presence is causing HSCs, thus threatening the survival of this emblematic rare carnivore in this region. Therefore, the increasing agriculture expansion, human population, and livestock population in and around KNP may also have implications for the management and conservation of the target species of KNP due to increasing HSCs.

China Pakistan Economic Corridor (CPEC) is a large infrastructure project including roadways, pipelines, and railways, many of which bisect the northern mountainous regions of Pakistan [79]. These areas of Northern Pakistan report 50% of the country's biodiversity, harboring endangered species including Siberian ibex, Himalayan brown bear, snow leopard, Himalayan black bear, and Himalayan ibex [80,81]. If not planned and implemented appropriately, CPEC project would impose pressing environmental challenges along the route as new roads and development projects open up previously inaccessible areas to logging, mining, land conversions, and other anthropogenic impacts. To shift these potential destructive patterns for protected areas and megafauna, previous studies have recommended to plan and support scientific ecological impact assessment, protected areas, wildlife corridors, conservation planning methods, and transboundary and international collaborative efforts in conservation [82]. The planned Karakoram Highway, aimed to increase economic and infrastructure development along the route, bisects prime snow leopard habitats as shown in this study. The potential impact of the highway on suitable habitats for snow leopards needs to be further investigated and monitored.

## 5. Conclusions

The current study is the first to determine the suitable snow leopard habitats using a MaxEnt model in its core habitat of northern Pakistan. The model reveals that snow leopards prefer comparatively lower elevations and that there is a clear overlap between suitable habitats for snow leopards and human activities. Only a fractional portion of this region is suitable snow leopard habitat and increasing human encroachment into this habitat has occurred mainly in the last decade, fuelled by the ever-growing human and livestock populations.

The overlap between suitable snow leopard habitat and human activities, leading to closer contact between humans and snow leopards, may be the reason for the increasing HSCs in this region. Implementing proper monitoring and conservation measures to manage and mitigate HSCs is crucial to maintaining this regional scale's snow leopard distribution and population size. More importantly, sound land use planning that considers

both wildlife needs and human demands and achieves a balance between them is critically important to mitigate potential HSCs in the long term.

**Supplementary Materials:** The following are available online at https://www.mdpi.com/article/10.3390/su132313229/s1, Figure S1: Response of Snow leopard (Panthera uncia) distribution to bio_2, bio_4, bio_7 and bio_8. Figure S2: Response of Snow leopard (Panthera uncia) distribution to bio_13, bio_15, bio_16 and slope. Figure S3: Response of Snow leopard (Panthera uncia) distribution to Land cover and aspect. Figure S4: The statistical graphs of the MaxEnt model output results: (a) the ROC (receiver operating characteristics) curve and the average test AUC (Area under Curve) for accuracy of the analysis of habitat prediction; (b) analysis of the omission test rate and predicted area. Figure S5: Current land use and land cover of the study area. Figure S6: Location of the sampling points obtained through the sign survey. Table S1: Analysis of correlation coefficient of the environmental variables. Table S2: Snow leopard presence points obtained from the study area.

**Author Contributions:** Conceptualization, W.R., J.S. and I.u.R.; methodology, B.A. and I.K.; software, F.Y.; validation, W.R. and J.S.; formal analysis, M.N.B. and F.Y.; investigation, W.R. and M.Q.; resources, J.S. and I.u.R.; data curation, E.B.; writing—original draft preparation, W.R. and J.S.; writing—review and editing, E.B.; visualization, I.K.; supervision, J.S. and I.u.R.; project administration, M.Q.; funding acquisition, J.S. All authors have read and agreed to the published version of the manuscript.

**Funding:** This study was supported by the National Natural Science Foundation of China (Project No. 31572281), China Scholarship Council (CSC# 201639180003), and the Interdisciplinary Research Funds of Beijing Normal University.

**Institutional Review Board Statement:** This study was approved by the School of Environment of the Beijing Normal University. All authors provided written informed consent. For the fieldwork, a No Objection Certificate (NOC SOH (FSC)-21/2016(NOC)) was obtained from the local government for the Pakistani Pamir region. This study does not involve wild animal capturing or handling.

**Informed Consent Statement:** Not applicable.

**Data Availability Statement:** The datasets generated during and/or analysed during the current study are available from the corresponding author on reasonable request.

**Acknowledgments:** We gratefully acknowledge the support provided by CAPRIL (Center for Applied Policy Research in Livestock), University of Veterinary and Animal Sciences, Lahore (Pakistan). We also acknowledge two anonymous reviewers for their constructive comments on an earlier version of this article.

**Conflicts of Interest:** The authors declare that they have no known potential sources of conflict of interest—nor competing financial interests or personal relationships—that could have appeared to influence the work reported in this paper.

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
