# Peer review of "Modelling Potential Distribution of Snow Leopards in Pamir, Northern Pakistan: Implications for Human–Snow Leopard Conflicts"

_sustainability, doi:10.3390/su132313229_

Round 1
Reviewer 1 Report
Page 1, line one of the introduction: uncia with lowercase.
Paragraph 1 of 2.3 section: correct the verbal time in redaction.
Show sampling points on a map.
The sampling period is very short (March to May 2018), the information collected does not allow to depict the distribution of the species in detail. The sampling period should at least represent two different seasons of the year. The cameras were only used for 20 days, generating an even greater bias with the rest of the sampling methods.
Describe global land cover variable used. Land cover change NDVI downloaded from the European Space Agency is a continuous map not categorical as described in table. If the authors use the described map in 2.4 section, should detail it in the appropriate section. They should also summarize how they combined the information from the NDVI and MNDWI to generate the categories.
Specify of the 85 records, which ones were obtained from the camera traps?
If the authors are using land use as a variable to model the distribution of the species, it is redundant to reuse the variable to determine overlap, if the species is preferring or avoiding modified habitats that is reflected by the influence of the variable in the model.
The collection data are almost 10 years different from the land use data used in the model. Justify its applicability.
Give more details about the supervised classification validation process.
The authors explain in the methodology how they responded to the objective: identify overlapping areas between the snow leopard distribution range and areas used by humans, which has implications for the management of HSCs.
They do not explain how analyzed the overlap of land use and model.
In the table 1 and 2 change the nomination of the variables to the name of the variable (not bio 02, etc.)
There are very few camera traps data that confirm the presence of the species.
Justify the applicability of the method to define the threshold for the case study.
In figure 2 generate your own graphs and do not place the screenshots obtained by default in the program.
Figure 5a should also show the model overlap with land use change.
Author Response
Reviewer #1:
- Page 1, line one of the introduction: uncia with lowercase.
Response: Thank you for your comment. The word is corrected now.
- Paragraph 1 of 2.3 section: correct the verbal time in redaction.
Response: Thank you for your comment. These sentences are now corrected.
- Show sampling points on a map.
Response: Thank you for your guidance. The points are now shown on the map (Figure S6) in the supplementary materials.
- The sampling period is very short (March to May 2018), the information collected does not allow to depict the distribution of the species in detail. The sampling period should at least represent two different seasons of the year. The cameras were only used for 20 days, generating an even greater bias with the rest of the sampling methods.
Response: Thank you for your comment. This sampling period was taken in the late winter and early spring (March to May 2018). Because Snow leopard is cryptic and elusive species. It is not easy to obtain the sampling points. However, during the mating season (in late winter and early spring in the high mountains), the Snow leopard extensively marks the area, and it is easy to get the sampling points.
- Describe global land cover variable used. Land cover change NDVI downloaded from the European Space Agency is a continuous map not categorical as described in table. If the authors use the described map in 2.4 section, should detail it in the appropriate section. They should also summarize how they combined the information from the NDVI and MNDWI to generate the categories.
Response: Thank you for your comment. We have now stated the technical details in detail.
- Specify of the 85 records, which ones were obtained from the camera traps?
Response: Thank you for your comment. The details about the records are given in the supplementary materials. The records obtained from camera traps are also highlighted and shown separately.
- If the authors are using land use as a variable to model the distribution of the species, it is redundant to reuse the variable to determine overlap, if the species is preferring or avoiding modified habitats that is reflected by the influence of the variable in the model.
Response: Thank you for your comment. This is indeed an issue worthy of attention. However, it should be noted that the resolution of the land cover data used for the MAXENT model is low in temporal and spatial, so it can only be used to judge extreme situations. It may be dispensable, although we choose to use it. As shown in Figure S2, in general, it has a low contribution. But in the analysis of land-use dynamics, we used self-made, more refined data.
- The collection data are almost 10 years different from the land use data used in the model. Justify its applicability.
Response: Thank you for your comment. But this worry may be unnecessary because the resolution of this data is 30 seconds, and it obviously cannot identify the fine changes in ten years. As mentioned in the previous reply, this coarse-resolution variable used in the model is just for the completeness of the modeling. Note that Figure 3 is the result of resolution optimization after resampling, which may cause misunderstandings.
- Give more details about the supervised classification validation process.
Response: Thank you for your comment. Same as the content 4, we now state the technical in details
- The authors explain in the methodology how they responded to the objective: identify overlapping areas between the snow leopard distribution range and areas used by humans, which has implications for the management of HSCs.
Response: Thank you for your comment.
- They do not explain how analyzed the overlap of land use and model.
Response: Thank you for your comment. A simple statement was added in the last sentence of section 2.4.
- In the table 1 and 2 change the nomination of the variables to the name of the variable (not bio 02, etc.)
Response: Thank you for your guidance to improve the manuscript. The nomination of the variables is now changed to the name of the variable in Table 2. However, table 1, shows the codes and the nomination of the variables for easy understanding of the readers.
- There are very few camera traps data that confirm the presence of the species.
Response: Thank you for your comment. The Snow leopard is a cryptic and elusive carnivore. It is also a rare species and has thin distribution in its habitat range. It is challenging to obtain the snow leopard camera presence. The camera trapping in the snow leopard range takes months, and the camera trapping stations did not have a single snow leopard photographed.
Furthermore, the single snow leopard range may be as comprehensive as 1000 square kilometers. So, it is not easy to have many snow leopard photographs in the snow leopard range. This is particularly difficult in these highest mountains of the world and the very remote terrain.
- Justify the applicability of the method to define the threshold for the case study.
Response: Thank you for your comment. As you know, the definition of this threshold is not absolute. Therefore, we refer to some existing publications, which have been cited in the second paragraph of 2.5.
- In figure 2 generate your own graphs and do not place the screenshots obtained by default in the program.
Response: Thank you for your comment. Figure 2 is now moved to the supplementary materials.
- Figure 5a should also show the model overlap with land use change.
Response: Thank you for your comment. We tried but gave up. Perhaps you have imagined how bad it looks. To make Figure5a (now Figure 3) redundant, we deleted Figure3 (now Figure 2).
We hope that this will be a final file. The uploaded file is fixed. Thank you so much for your kind comments.
We wish to thank the reviewers for their detailed review, efforts, and contribution to the scientific community as a reviewer.
Sincerely yours,
Dr. Jianbin Shi
Professor
jbshi@bnu.edu.cn
Contact number: +86 13911 323726

Reviewer 2 Report
In this study, a MaxEnt model predicted the snow leopard’s potential distribution and analyzed the land use/cover to determine the habitat interactions of snow leopards’ with human/livestock in Karakoram-Pamir, northern Pakistan. The paper is well organized, has clear objectives and the drawn conclusions are coherent with the obtained results. This model can be used in the conservation action for this species in Pakistan.
The key-words should be alphabetically arranged
It is Human-Snow leopard Conflicts (HSC)
It is Species Distribution Models (SDMs)
I think that you should add this recent references as example to support your sentence “MaxEnt is one of the most widely used SDMs for predicting the potential distribution of a given species [34-36]”. I would like to suggest:
Di Pasquale, G., Saracino, A., Bosso, L., Russo, D., Moroni, A., Bonanomi, G., & Allevato, E. (2020). Coastal pine-oak glacial refugia in the Mediterranean basin: A biogeographic approach based on charcoal analysis and spatial modelling. Forests, 11(6), 673.
Zhao, Y., Deng, X., Xiang, W., Chen, L., & Ouyang, S. (2021). Predicting potential suitable habitats of Chinese fir under current and future climatic scenarios based on Maxent model. Ecological Informatics, 64, 101393.
Did you have analyse your data for spatial autocorrelation ?
https://pro.arcgis.com/en/pro-app/latest/tool-reference/spatial-statistics/spatial-autocorrelation.htm
Table 1 should be moved in the supplementary materials
I think that you should add this recent references as example to support your sentence “Maxent requires a threshold value to distinguish suitable from unsuitable habitats”. I would like to suggest:
Ancillotto, L., Mori, E., Bosso, L., Agnelli, P., & Russo, D. (2019). The Balkan long-eared bat (Plecotus kolombatovici) occurs in Italy-first confirmed record and potential distribution. Mammalian Biology, 96(1), 61-67.
Su, H., Bista, M., & Li, M. (2021). Mapping habitat suitability for Asiatic black bear and red panda in Makalu Barun National Park of Nepal from Maxent and GARP models. Scientific Reports, 11(1), 1-14.
Figure 2 should be moved in the supplementary materials
Figure 4 should be moved in the supplementary materials
I think that you should add this recent references as example to support your sentence “These areas of Northern Pakistan report 50% of the biodiversity of the country, harboring endangered species including Siberian Ibex, Himalayan brown bear, Snow leopard, Himalayan black bear, Himalayan Ibex”. I would like to suggest:
Goursi, U. H., Anwar, M., Bosso, L., Nawaz, M. A., & Kabir, M. (2021). Spatial distribution of the threatened Asiatic black bear in northern Pakistan. Ursus, 2021(32e13), 1-5.
Author Response
Reviewer #2:
- The key-words should be alphabetically arranged
Response: Thanks for your comment. The keywords are arranged alphabetically now.
- It is Human-Snow leopard Conflicts (HSC)
Response: Thank you for guiding us to improve the paper. The required correction is done in the revised version.
- It is Species Distribution Models (SDMs)
Response: Thank you for bringing up this mistake. The corrections are made in the manuscript now.
- I think that you should add this recent references as example to support your sentence “MaxEnt is one of the most widely used SDMs for predicting the potential distribution of a given species [34-36]”. I would like to suggest:
Di Pasquale, G., Saracino, A., Bosso, L., Russo, D., Moroni, A., Bonanomi, G., & Allevato, E. (2020). Coastal pine-oak glacial refugia in the Mediterranean basin: A biogeographic approach based on charcoal analysis and spatial modelling. Forests, 11(6), 673.
Zhao, Y., Deng, X., Xiang, W., Chen, L., & Ouyang, S. (2021). Predicting potential suitable habitats of Chinese fir under current and future climatic scenarios based on Maxent model. Ecological Informatics, 64, 101393.
Response: Thank you for your good comments for improving the manuscript. The references are now added and revised accordingly.
- Did you have analyse your data for spatial autocorrelation?
https://pro.arcgis.com/en/pro-app/latest/tool-reference/spatial-statistics/spatial-autocorrelation.htm
Response: Thank you for your guidance. The correlation is done and analysis of correlation coefficient of the environmental variables is given in the supplementary materials. A separate document is attached now to the Supplementary materials.
- I think that you should add this recent references as example to support your sentence “Maxent requires a threshold value to distinguish suitable from unsuitable habitats”. I would like to suggest:
Ancillotto, L., Mori, E., Bosso, L., Agnelli, P., & Russo, D. (2019). The Balkan long-eared bat (Plecotus kolombatovici) occurs in Italy-first confirmed record and potential distribution. Mammalian Biology, 96(1), 61-67.
Su, H., Bista, M., & Li, M. (2021). Mapping habitat suitability for Asiatic black bear and red panda in Makalu Barun National Park of Nepal from Maxent and GARP models. Scientific Reports, 11(1), 1-14.
Response: Thank you for your guidance. We have added these recent references to support the sentences
- Figure 2 should be moved in the supplementary materials
Response: Thank you for your comment. The Figure 2 is now moved to the supplementary materials.
- Figure 4 should be moved in the supplementary materials
Response: Thank you for guiding to improve the manuscript. The Figure 4 is now moved to the supplementary materials.
- I think that you should add this recent references as example to support your sentence “These areas of Northern Pakistan report 50% of the biodiversity of the country, harboring endangered species including Siberian Ibex, Himalayan brown bear, Snow leopard, Himalayan black bear, Himalayan Ibex”. I would like to suggest:
Goursi, U. H., Anwar, M., Bosso, L., Nawaz, M. A., & Kabir, M. (2021). Spatial distribution of the threatened Asiatic black bear in northern Pakistan. Ursus, 2021(32e13), 1-5.
Response: We appreciate the reviewer's comment on this. The above recent reference is added to the manuscript now.
We hope that this will be a final file. The uploaded file is fixed. Thank you so much for your kind comments.
We wish to thank the reviewers for their detailed review, efforts, and contribution to the scientific community as a reviewer.
Sincerely yours,
Dr. Jianbin Shi
Professor
jbshi@bnu.edu.cn
Contact number: +86 13911 323726

Round 2
Reviewer 1 Report
The observations were attended.
Reviewer 2 Report
Well done!